# Brief communication: Characteristic properties of extreme wave events observed in the northern Baltic Proper, Baltic Sea

Jan-Victor Björkqvist[1], Laura Tuomi[1], Niko Tollman[1], Antti Kangas[1], Heidi Pettersson[1], Riikka Marjamaa[1], Hannu Jokinen[1], and Carl Fortelius[1]

[1]Finnish Meteorological Institute, Erik Palménin aukio 1, P.O.Box 503, FI-00101 Helsinki, Finland

*Correspondence to:* Jan-Victor Björkqvist (jan-victor.bjorkqvist@fmi.fi)

**Abstract.** A significant wave height of 7 m has been measured five times by the northern Baltic Proper wave buoy in the Baltic Sea, exceeding 8 m twice (2004 & 2017). We classified these storms into two groups by duration and wave steepness. Interestingly, the two highest events exhibited opposite properties, with the 2017 event being the longest storm on record. This storm is also the first where the harshest wave conditions were modelled to occur in the western part of the Baltic Proper. The metrics quantifying the storm's duration and steepness might aid in issuing warnings for extreme wave conditions.

## 1 Introduction

Extreme wave conditions impact the transport and safety at sea. They slow down larger vessels and can threaten the safety of smaller ships. Both operational wave measurements and wave forecasting models are needed to issue warnings and provide accurate estimates of the conditions seafarers will face along their routes.

The wind conditions play a key part in the formation of the extreme wave conditions at sea. Because of the small size and geometry of the Baltic Sea, both the fetch and the duration of the wind event limit the wave growth. The Baltic Proper has the largest fetch and the harshest wave conditions of all the Baltic Sea sub-basins (Tuomi et al., 2011). Earlier studies have shown that the highest waves are typically in the north- and southeastern part of the domain (e.g. Jönsson et al., 2003; Tuomi et al., 2011). The highest wave event on record happened when the northern Baltic Proper (NBP) wave buoy measured a significant wave height of 8.2 m in December 2004. Recently, in January 2017, an 8 m significant wave height was recorded for the second time in the 20 year long measurement history at the same location.

Even higher waves have been estimated to have occurred in the northern Baltic Proper during the wind storm Gudrun in January 2005. The highest waves were evaluate to be slightly south-southeast from the location of the NBP wave buoy, where a significant wave height of 7.2 m was measured. Wave experts who reviewed the results of three validated wave forecast models and the wind conditions during the storm concluded that the highest significant wave height was in the order of 9.5 m (Soomere et al., 2008).

The ship routes from Stockholm to Helsinki and Tallinn – the capitals of Sweden, Finland and Estonia – cross the area where the highest waves occur. Furthermore, the wave direction in this area is typically from south or southwest during storms, thus propagating perpendicular to the shipping routes. The most disastrous accident along these routes was the sinking of MS

*Estonia* in 1994; 852 lives were lost. The significant wave height was estimated to be between 4–5 m during the accident (Joint Accident Investigation Commission of Estonia and Sweden, 1997). While the wave conditions were evaluated to not be the primary reason behind this accident, they caused damage to the vessel and complicated the rescue missions.

In this paper we evaluate the characteristic properties of five extreme wave events in the Baltic Sea measurement records. A special attention is given to the two storms: "Rafael" in 2004 and "Toini" in 2017. The accuracy of the wave forecast of the Finnish Meteorological Institute (FMI) is evaluated and the issuing of warnings for extreme events is briefly discussed based on the findings.

## 2   Description of the area and the available data

The Baltic Sea is a semi-enclosed water body with several sub-basins (Fig. 1a). Since this paper focuses on extreme wave events, we will limit our study to the largest basin – the Baltic Proper.

To analyse the storms we use wave measurements from the operational wave buoy of the Finnish Meteorological Institute (FMI) that is moored at a depth of 100 m in the northern Baltic Proper (NBP). The data from another Directional Waverider moored on the eastern side of the Swedish island of Gotland provides some spatial information about the wave conditions. To evaluate the wind conditions we use 10-min average wind data from FMI's weather station at Bogskär. An overview of the locations can be found in Fig. 1a.

We use the parameters significant wave height ($H_s = H_{m_0}$) and peak wave period ($T_p$) (Dat, 2017) in the analysis. The mean inverse wave steepness $\langle \lambda_p / H_s \rangle$ serves as an indicator of the steepness conditions, where the peak wavelength $\lambda_p$ is estimated from the peak period using linear wave theory and the brackets denote the temporal average.

We analyse the spatial attributes of the wave field using FMI's operational wave forecast model WAM cycle 4 (WAMDIG, 1988; Komen et al., 1994). WAM is a third generation phase averaged spectral wave model that solves the action balance equation to simulate the wave energy at each grid point. This wave model has been sucessfully implemented to the Baltic Sea (e.g. Tuomi, 2008; Räämet and Soomere, 2010; Tuomi et al., 2011). In 2004 FMI's operational wave forecast model had a spatial resolution of 22 km and output time interval of 3 h. The spatial resolution has been increased and is currently 4 nmi ($\sim$ 7.4 km), while the output temporal resolution is 1 h.

The surface wind field at 10 m height from FMI's operational numerical weather predictions system HIRLAM (HIRLAM-B, 2017) function as the meteorological forcing for the wave model. The present FMI-HIRLAM has 0.068 ° horizontal resolution and 64 vertical terrain-following hybrid levels. The 54 hour forecasts are run four times a day (00, 06, 12, and 18 UTC) using boundary conditions from the Boundary Condition Optional Project of the ECMWF (European Centre for Medium-Range Weather Forecasts). In 2004 FMI's NWP system HIRLAM had 0.2 degrees horizontal resolution and 40 vertical levels. The physics and the parameterisations in HIRLAM has also improved over the years, which has increased the accuracy of the forecast surface winds (Eerola, 2013).

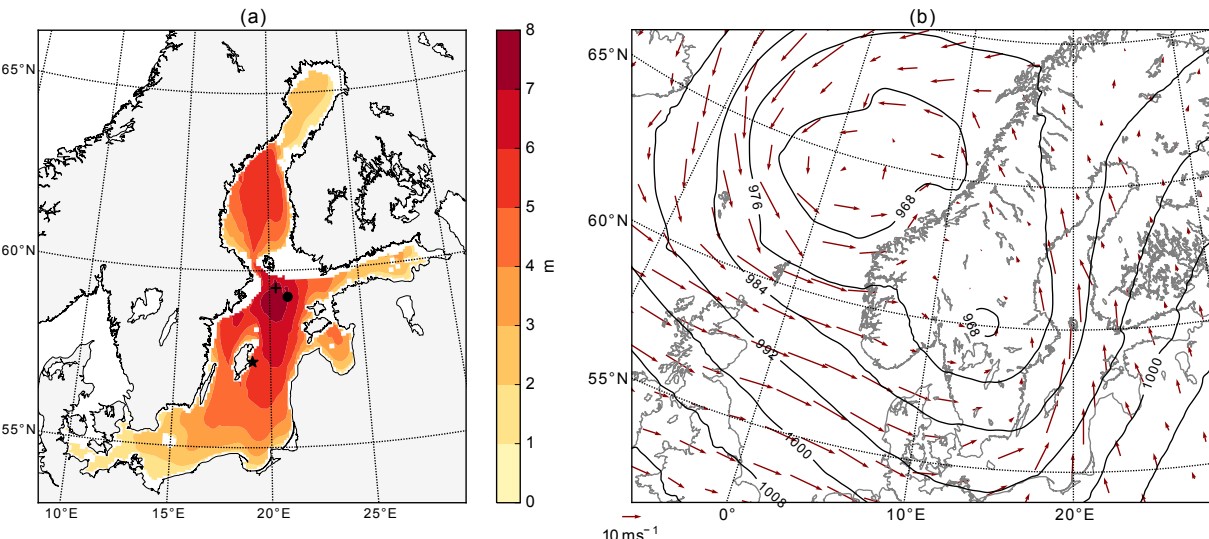

**Figure 1.** The modelled significant wave height at 22:00 UTC **(a)** and the meteorological conditions at 21:00 UTC **(b)** during storm Toini in 11 January 2017. The locations of the NBP wave buoy (circle), the Gotland wave buoy (star) and Bogskär wind measurements (plus) are also shown in **(a)**.

## 3 Characteristic storm properties

For the purpose of this paper a storm is defined as an event when the significant wave height exceeds 7 m at least once. We further define the duration of a storm as the time the significant wave height exceeds 6 m. In the measurement history of the NBP wave buoy (1996–2017) this amounts to five storm events: two in December 1999, one in December 2004, one in January 5 2005 and one in January 2017.

The NBP wave buoy has measured a significant wave height of 8 m only twice (2004 and 2017). During the other three storms the measured maximum has been under 7.5 m (Table 1). The measured maximum values of the peak wave period $T_p$ in four of the five storms were 13 s. The observed peak period during the first storm in 1999 (henceforth 1999a) didn't exceed 12 s. However, the peak period was still growing at the start of an unfortunate three hour gap in the measurements.

10 Based on a 6 year model hindcast (November 2001 to October 2007) Tuomi et al. (2011) found the statistical exceedance time for a significant wave height of 6 m to be 8.8 h per year at the NBP wave buoy. The analysis of the storms reveals that the true duration of the storms have been slightly longer, typically around 10–15 h (Table 1).

A comparison of the two most sever storms (Rafael in 2004 and Toini in 2017) reveals several characterising differences. Rafael was short, with a 6 m exceedance time of only 9 h, while Toini lasted 6.5 h longer. The mean inverse significant 15 steepness were 27 and 30 for Rafael and Toini respectively, meaning that the waves were steeper during Rafael. The difference in steepness is partially explained by the behaviour of the peak period. It reached its maximum value during the storm in 2017, while the maximum peak period in 2004 was observed after the significant wave height had decayed to under 6 m (not shown).

**Table 1.** The maximum values of the wave parameters during the storms. The exceedance time for the significant wave height over 6 m and mean inverse significant steepness for that exceedance time is also given.

| Time | max $H_s$ | max $T_p$ | $H_s \geq 6$ m | $\langle \lambda_p / H_s \rangle$ |
|---|---|---|---|---|
| 06 December 1999 | 7.4 m | 12.0 s | $< 7$ h | 25 |
| 17 December 1999 | 7.4 m | 12.5 s | 13.5 h | 30 |
| 22 December 2004 | 8.2 m | 12.7 s | 9.0 h | 27 |
| 09 January 2005 | 7.2 m | 12.8 s | 13.5 h | 29 |
| 11 January 2017 | 8.0 m | 12.5 s | 15.5 h | 30 |

Also other storms from 1999 and 2005 can be classified to one of the two groups set by the 2004 and 2017 events. One group is identified by a short duration, late occurrence of the maximum peak period and steeper wave conditions (1999a, 2004). The second group consists of longer storms that reach their maximum peak period during the 6 m exceedance time, resulting in less steep wave conditions (1999b, 2005, 2017).

The wave observations from the NBP cannot be considered entirely representable for the entire Baltic Proper. The highest modelled wave events have been placed either south-southeast of the wave buoy during Gudrun in 2005 (Soomere et al., 2008), slightly west of the wave buoy during Toini in 2017 (Fig. 1), or slightly east of the wave buoy during Rafael in 2004 (not shown). High waves have also been modelled in the southern Baltic Sea (e.g. Jönsson et al., 2003), which is an area suffering from an acute lack of wave measurements. However, the sparsity of remotely sensed wave data and the uncertainties related to modelling the wave extremes (Fig. 2) underlines the usability of the reliable long term wave buoy measurements presented in this paper.

## 4 Forecasting

### 4.1 Toini 2017

On 10–12 January a vast low pressure was situated over the Norwegian Sea while a deepening secondary low formed over southern Scandinavia (see Fig. 1b). The secondary low moved northwards along the east coast of Sweden. This weather pattern created circumstances where southerly wind was in gale or strong gale force approximately 20 hours in the entire Baltic Proper, while the variation in wind direction was insignificant.

Toini was forecasted quite well already 24 h before the observed maximum (Fig. 2a). The biggest difference is that the forecasts available 18 h and 24 h prior to the storm predicted the maximum significant wave height to take place at 02:00 UTC, while the forecasts available 6 h before and during the storm predicted the maximum at 23:00 UTC and 22:00 UTC respectively. The observed maximum occurred at 22:30 UTC. The storm duration was also predicted more correctly closer to the storm, with a 9 h duration 24 h before the storm compared to a 13 h duration 6 h prior the the event. The maximum

significant wave height was nevertheless underestimated in all forecasts. The model bias for the 6 m exceedance time ranged from -0.5 m to -0.8 m in the different forecasts.

The predicted mean inverse significant steepness for the 6 m exceedance time was 29 for all the lead times, providing an accurate description of the steepness conditions. The peak period was predicted correctly in the sense that it reached its maximum value during the storm period, just as observed. The values of the peak period were underestimated by roughly 1 s (not shown). The modelled peak period did not exceed 12 s anywhere in the Baltic Proper.

In the forecast available 24 h prior to the storm the highest significant wave height was 7.0 m slightly south-west of the wave buoy at 22:00 UTC. In the forecast available during the storm the maximum (7.4 m) was located west of the wave buoy. The most extreme wave events have, up until now, been modelled to take place in the eastern part of the Baltic Proper (Tuomi et al., 2011). The exceptional wave conditions in the western part of the Baltic Proper during Toini were also captured by the Gotland wave buoy, which measured it's highest significant wave height to date (5.6 m).

## 4.2   Rafael 2004

On 21–22 December 2004 two low pressure centers over the Arctic Ocean and the North Atlantic joined together to form a strong low pressure system with a single center northwest of Norway. The south to southwest wind speed increased to storm force in the northern Baltic Proper. Compared to Toini the duration of strong gale winds was much shorter in Rafael, lasting roughly 8 hours. The wind direction was also more southwesterly compared to more southerly wind during Toini.

The forecast of Rafael underpredicted the significant wave height by up to 0.9 m and the peak period by about 2 s in all forecasts available less than 24 h before the storm. The model bias for the 6 m exceedance time ranged from -0.7 m to -1.1 m in the different forecasts. As for Toini, the predicted time of the extreme values differ between the forecasts (Fig. 2b). The maximum significant wave height was predicted correctly at 21:00 UTC in the forecasts available less than 12 h prior to the event. The observed maximum occurred at 20:00 UTC.

The length of the storm in the forecasts was between 3 h and 6 h, which is shorter than the observed 9 h duration. We can conclude that the duration was underestimated in the forecast (Fig. 2b), even though the coarse 3 h time resolution of the model output makes it challenging to quantify the exact duration. The forecasted steepness values between 24 and 27 were in good accord with the observed value of 27, which was also exactly the value for the forecast available 24 h before the storm. The maximum modelled significant wave height for the entire Baltic Proper basin was 7.5 m in the latest forecast.

## 5   Warning of extreme waves

Warnings for severe and extreme wave conditions were launched at FMI in 2011. The wave warnings are issued for ships and boats together with other meteorological warnings regularly 7 times a day, or as needed in case of an unexpected situation. The thresholds for the warnings are 2.5, 4 and 7 m in significant wave height. The first 2.5 m limit is important for smaller boats, especially during the leisure boating season, while the 4 m limit represents wave conditions that might impact even larger vessels. The 7 m significant wave height is considered to be potentially dangerous for all ships.

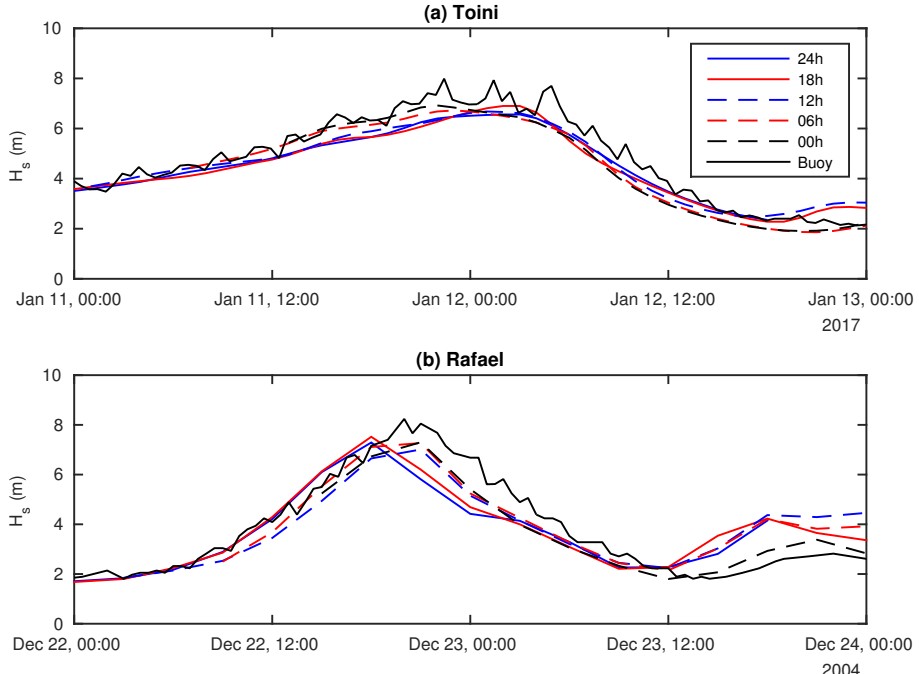

**Figure 2.** Forecasts for the significant wave height $H_s$ for storms Toini **(a)** and Rafael **(b)**. The notation "12h" means that the forecast was available 12 hours prior to the observed maximum. The continuous black line is the values measured by the NBP wave buoy.

Since 2011, Toini is the first storm in the Baltic Sea when the significant wave height has exceeded 7 m. However, for Toini the warning was given only for severe wave conditions, with a more specific estimate of 6–7 m for the northern Baltic Sea. Although an extreme wave warning was considered, the wave forecast 24 h before the storm predicted the highest significant wave height to be 6.95 m. The expert estimate for the significant wave height based on an analysis of meteorological and oceanographic forecasts and statistics was 6.9 m. The warning was updated to extreme wave conditions during the storm as the observed significant wave height exceeded 7 m.

The accuracy of the wave forecasts is constantly evaluated against the wave buoy measurements. The verification results show that FMI's wave forecast system has good accuracy at the NBP buoy location with a slight tendency to underestimate the largest values of significant wave height. However, both the NWP systems and wave forecast models are regularly upgraded, resulting in new combinations of e.g. spatial and temporal resolutions, physics, and parametrisations (Tuomi, 2008; Eerola, 2013). Although the less extreme values are known to be modelled well (Tuomi, 2008; Tuomi et al., 2011), it is challenging to obtain a comprehensive understanding of how the operational models behave in extreme circumstances, since significant wave heights of over 7 m occur very rarely.

A discussion concerning the issuing of wave warnings for the Baltic Sea should be initiated between the relevant institutes and end users. In addition to re-establishing and harmonising the thresholds of significant wave heights, the use of other parameters (e.g. duration) should also be explored in light of the difficulties of predicting a single maximum value for the wave

height. Any decision to include new parameters should be based on the needs of the seafarers. On a more general note, the use of ensemble forecasts might prove useful when issuing wave warnings. An in-depth study is nevertheless needed in order to quantify to which extent the added information warrants the increased computational cost.

## 6  Summary

We analysed the five wave events in the Baltic Sea that have exceeded a significant wave height of 7 m during 1996–2017. In addition to the maximum wave height we calculated the duration ($H_s > 6$ m) and the mean inverse wave steepness for the storm. On the basis of our analysis we classify the extreme wave events into two groups. One category is characterised by a long duration (> 10 h) and a high mean inverse significant steepness (> 28). The other group consists of shorter and steeper storm events (see Table 1).

The two storms with the highest significant wave heights (8.2 m in 2004 and 8.0 m in 2017) exhibited different characteristics. Toini in 2017 had the longest duration (15.5 h) to date, but it is also the first storm where the wave model places the most extreme wave conditions in the western part of the northern Baltic Proper. Rafael in 2004 remains the most extreme event if classified solely based on the maximum observed significant wave height.

The duration and steepness characteristics of Toini were fairly well resolved by the wave forecasts. These metrics may therefore provide an additional tool to aid in deciding when to issue warnings for extreme wave conditions in the future.

## 7  Data availability

The measurement time series for all storm events and the time series for the forecasts from 2004 and 2017 are available as supplementary material to this article.

*Author contributions.*  The paper was initiated by Björkqvist and Tuomi. The wave measurements were analysed by Björkqvist, Pettersson and Jokinen, while the wave model data was reviewed by Björkqvist, Marjamaa, Kangas and Tuomi. Tollman, Kangas and Fortelius were responsible for the analysis of the meteorological conditions. The manuscript was prepared by Björkqvist and Tuomi with contributions from all co-authors.

*Acknowledgements.*  We would like to thank the reviewers Dr. Alvaro Semedo and Dr. David Moncoulon for their comments and suggestions, which enabled us to improve our article. This work is partly funded by the SmartSea project of the Strategic Research Council of Academy of Finland, grant No: 292 985. We gratefully acknowledge the valuable comments provided by Havu Pellikka.

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
