# Peer review of "Brief communication: Characteristic properties of extreme wave events in the Baltic Sea"

_Natural Hazards and Earth System Sciences, 2017_

## Short Comment (SC1) · 6 Jun 2017

Review of the Natural and Hazards Earth System Sciences - Discussions: "Brief communication: Characteristic properties of extreme wave events in the Baltic Sea" by Björkqvist et al. (Rev nhess-2017-117)

An insight on the several storm case studies (extreme wave height events) has been presented, having in mind the "characteristic properties of extreme wave events in the Baltic Sea" is presented by the authors. This (short) paper presents a relatively detailed study of several storms which had a strong effect in extreme wave height events. The study is directed not to a climatological perspective (as the title might insinuate), but more to the analyse of the storms having in mind the operational forecast scores. The

study is simple (simplistic, to a certain extent), and could have been explored into a greater detail. Nevertheless, it is well written, and has a great utility to the seafarers and to the operational forecasters in the area.

The title might direct the readers to a climactic study, which is not the case, since the detailed characteristics of extreme waves in the Baltic Sea are not presented. I would like the authors to defend their point of view on this matter.

The manuscript is, in general, well written, and the ideas are well presented and well defended. Nevertheless, it lacks depth, which can be explained by the "short communication" format. Nothing against. Just that this subject and idea deserves a more detailed analysis.

Here and there some references to back some statements are needed. Some suggestions are made below, but I challenge the authors read the paper again and make their own review on this matter.

Minor comments and suggestions on the text:

P1, L9 – Replace "in" with "along". P1, L10 – extreme conditions of what? P1, L12: Add reference after sub-basins P1, L17 – estimated? Modelled, maybe. P2, L21 - The present resolution of the WAM setup in the FMI operational wave product is 4 nautical miles? Since this is not a very common scale (unit) maybe it should be explained. Replace "timestep" with "temporal resolution". P2, L28 – affects how. I presume it improves. P3, L3-4 – Sentence starting with "Of the. . ." is confusing. Re-write. P3, L15 – Erase "the" before "other". P4, L3 – What is a "vast low pressure"? This sentence is out of context. P4, L10 – Replace "was" after "maximum" with "occurred at". P4, L13 – How come mean? P5, L3 – Add "speed" after "wind". P5, L9 – Replace "was" with "occurred". P5, L15 – merge this sentence with the previous paragraph.

---

## Referee Comment (RC1) · A. Semedo (Referee) · 8 Jun 2017

Review comments of the Natural and Hazards Earth System Sciences – Discussions manuscript: "Brief communication: Characteristic properties of extreme wave events in the Baltic Sea" by Björkqvist et al.

An insight on the several storm case studies (extreme wave height events) has been presented, having in mind the "characteristic properties of extreme wave events in the Baltic Sea" is presented by the authors. This (short) manuscript presents a relatively detailed study of several Baltic Sea storms, which had a strong effect in extreme wave height events. The study is directed not to a climatological perspective (as the title might insinuate), but more to the analyse of the storms, having in mind the operational

forecast scores. The study is simple (simplistic, to a certain extent), and could have been explored into a greater detail. Nevertheless, it is well written, and has a great utility to the seafarers and to the operational forecasters in the area.

The title might direct the readers to a climactic study, which is not the case, since the detailed characteristics of extreme waves in the Baltic Sea are not presented. I would like the authors to defend their point of view on this matter.

The manuscript is, in general, well written, and the ideas are well presented and well defended. Nevertheless, it lacks depth, which can be explained by the "short communication" format. Nothing against. Just that this subject and idea deserves a more detailed analysis.

Here and there some references to back some statements are needed. Some suggestions are made below, but I challenge the authors to read the paper again and make their own review on this matter.

Minor comments and suggestions on the text:

P1, L9 – Replace "in" with "along". P1, L10 – extreme conditions of what? P1, L12: Add reference after sub-basins P1, L17 – estimated? Modelled, maybe. P2, L21 - The present resolution of the WAM setup in the FMI operational wave product is 4 nautical miles? Since this is not a very common scale (unit) maybe it should be explained. Replace "timestep" with "temporal resolution". P2, L28 – affects how. I presume it improves. P3, L3-4 – Sentence starting with "Of the..." is confusing. Re-write. P3, L15 – Erase "the" before "other". P4, L3 – What is a "vast low pressure"? This sentence is out of context. P4, L10 – Replace "was" after "maximum" with "occurred at". P4, L13 – How come mean? P5, L3 – Add "speed" after "wind". P5, L9 – Replace "was" with "occurred". P5, L15 – merge this sentence with the previous paragraph.

---

## Referee Comment (RC2) · D. Moncoulon (Referee) · 17 Jul 2017

This paper presents an analysis of some extreme wave event that occurred in the Baltic Sea and a comparison between measured data from the wave buoy and the forecasting system. As the introduction, which is well written and easy to read, the whole paper is interesting and in good English. The method is well explained when it is about wave measurements. The wind speed data could have been more detailed: is it gust wind or average wind speed? The forecast modeling is presented as an input data for comparison with measurements, and one might need more explanations about the wave model. This paper is about extreme wave events, but in paragraph 5 it is explained that wave heights of 2,5, 4 and 7m are significant for boats. This paper could have been improved by analyzing the forecast model for smaller wave

heights than 7m. In the introduction, it has been highlighted than during the accident of the MS Estonia, 4-5m wave height has been measured. Is the forecast system more accurate for smaller events which are probably more frequent? In the same idea, I also want to make the observation that some proposals to improve the forecast system could have been welcome as an opening in the conclusion of paragraph 5. In the forecasting paragraph, the comparison between model results and measurements could have been improved by the use of objective indicators (Nash criteria? RMSE ?). The forecast models are compared with a single station for wave parameters. Is this station fully representative of the heterogeneity of the Baltic Sea waves? This point should have been discussed. In conclusion, this small paper is in good format for the "brief communication" and should only be enhanced by taking into account these remarks to improve the scientific discussion.

---

## Author Comment (AC1) · 22 Aug 2017

Response to review comments by A. Semedo

Thank you for the informative and constructive comments. We will now answer the comments point by point.

**General comments**

*"The title might direct the readers to a climactic study, which is not the case, since the detailed characteristics of extreme waves in the Baltic Sea are not presented. I would*

*like the authors to defend their point of view on this matter."*

This is a fair point, since the title doesn't accurately reflect that the analysis is based (mainly) on point observations from one location. The title has been changed to "Brief communication: Characteristic properties of extreme wave events observed in the northern Baltic Proper, Baltic Sea".

*"The manuscript is, in general, well written, and the ideas are well presented and well defended. Nevertheless, it lacks depth, which can be explained by the "short communication" format. Nothing against. Just that this subject and idea deserves a more detailed analysis."*

It is true that the format of the "brief communication" limits how deeply the performance of the operational model can be validated. We chose this format since it seemed like the most suitable and efficient format for reporting the findings inspired by the current storm Toini. We had two main reasons for reporting the findings: 1) the storm Toini is interesting in regards to the location of the maximum wave heights and it's long duration, 2) the storm generated wide interest both in Finnish and Swedish media. However, the main interest in the general public was almost solely on "the height of the single highest wave". The somewhat "simplistic" nature of the work is a result of it being inspired by the popular interest and our willingness to be able to communicate information about storm events in a more sophisticated, yet understandable, way. We hope that the work presented in this paper can serve as material for a discussion about wave warnings in enclosed seas, especially in the Baltic Sea.

*"Here and there some references to back some statements are needed. Some suggestions are made below, but I challenge the authors to read the paper again and make their own review on this matter."*

We will add references to statements concerning to model performance and previously obtained results about wave conditions. Added references are Jönsson et al. (2003), Tuomi (2008), Räämet Soomere (2010) and Eerola (2013).

**Minor comments and suggestions on the text**

*"P1, L9 – Replace "in" with "along""*

This will be replaced

*"P1, L10 – extreme conditions of what?"*

Will add the word "wave" to clarify that the sentence is regarding extreme wave conditions.

*"P1, L12:Add reference after sub-basins"*

We will a reference to Tuomi et al. (2011), since it is one previous study to support this assertion. We will also add other references to the next sentence starting "earlier studies have shown...".

*"P1, L17 – estimated? Modelled, maybe."*

This is the estimate that was made by Soomere et al. (2008). The authors used both model data and wave measurements to produce a best estimate of the highest significant wave height.

*"P2, L21 – The present resolution of the WAM setup in the FMI operational wave product is 4 nautical miles? Since this is not a very common scale (unit) maybe it should be explained."*

Will add the resolution in km also.

*"Replace "timestep" with "temporal resolution". "*

Will replace this in the text.

*"P2, L28 – affects how. I presume it improves."*

We will change the text to indicate that the accuracy has been increased and add a

reference to Eerola (2013).

*"P3, L3-4 – Sentence starting with "Of the ..." is confusing. Re-write."*

This sentence can be rewritten as:

The NBP wave buoy has measured a significant wave height of 8 m only twice (2004 and 2017). During the other three storms the measured maximum has been under 7.5 m (Table 1).

*"P3, L15 – Erase "the" before "other"."*

Will erase the word "the"

*"P4, L3 – What is a "vast low pressure"? This sentence is out of context."*

"Vast" was used in the meaning that is covered a large area. However, we take the point that is is perhaps not well defined, and we removed the word "vast", since it's not necessary. The sentence is a part of the description of the atmospheric conditions during the storm. We can rewrite the two sentences to make this clearer and feel it will then be in context:

"On 10–12 January a low pressure area was situated over the Norwegian Sea while a deepening secondary low formed over southern Scandinavia (see Fig. 1). The secondary low moved northwards along the east coast of Sweden."

*"P4, L10 – Replace "was" after "maximum" with "occurred at"."*

This will be replaced in the text.

*"P4, L13 – How come mean?"*

The reported steepness is a mean steepness in the sense that it is calculated as a temporal mean. It is defined on P1, L16-18. We will add the information to the text that it is the mean calculated for the 6 meter exceedance time. We will also clarify that the sentence describes to predicted value.

*"P5, L3 – Add "speed" after "wind"."*

This will be added.

*"P5, L9 – Replace "was" with "occurred"."*

This will be replaced in the text.

*"P5, L15 – merge this sentence with the previous paragraph."*

This will be merged in the text.

**Other changes**

Will add that the measurement history is 20 years, P1, L15.

Will correct "east-southeast" to "south-southeast", P1, L18.

Will correct the word "Rafel" to "Rafael" P2, L5.

Will add that Bogskär is an FMI weather station, P2, L14.

**New references**

Jönsson, A., Broman, B., and Rahm, L.: Variations in the Baltic Sea wave fields, Ocean Eng., 30, 107 – 126, doi:http://dx.doi.org/10.1016/S0029-8018(01)00103-2, 2003.

Tuomi, L.: The accuracy of FIMR wave forecasts in 2002-2005, MERI – Report Series of the Finnish Institute of Marine Research, 63, 7–17, 2008.

Räämet, A. and Soomere, T.: The wave climate and its seasonal variability in the northeastern Baltic Sea, Estonian J. Earth Sci., 59(1), 100–113,

doi:10.3176/earth.2010.1.08, 2010.

Eerola, K.: Twenty-One Years of Verification from the HIRLAM NWP System, Weather and Forecasting, 28, 270–285, doi:10.1175/WAF-D-12-00068.1, 2013.

---

## Author Comment (AC2) · 22 Aug 2017

We appreciate that you took the time to review our manuscript. We will now answer the comments point by point.

*"The wind speed data could have been more detailed: is it gust wind or average wind speed?"*

We will add the information that it is the 10-min average wind speed that is being used.

*"The forecast modeling is presented as an input data for comparison with measurements, and one might need more explanations about the wave model."*

We will add a short explanation to the text and also added a few references to studies

where WAM has been implemented to the Baltic Sea. Will also add the information, that we have used WAM cycle 4. The performance of the model implementation to the Baltic Sea has been documented in the references.

"WAM is a third generation phase averaged spectral wave model that solves the action balance equation to simulate the wave energy at each grid point. This wave model has been sucessfully implemented to the Baltic Sea (e.g. Tuomi 2008; Tuomi et al., 2011)."

*"This paper is about extreme wave events, but in paragraph 5 it is explained that wave heights of 2,5, 4 and 7m are significant for boats. This paper could have been improved by analyzing the forecast model for smaller wave heights than 7m. In the introduction, it has been highlighted than during the accident of the MS Estonia, 4-5m wave height has been measured. Is the forecast system more accurate for smaller events which are probably more frequent¿'*

This is a good point. A more extensive validation of the wave forecast would indeed be interesting. Unfortunately, we could not fit a full validation to the format of the "brief communication". However, the performance of the wave model has been evaluated in previous studies (Tuomi 2008; Tuomi et al., 2011). Tuomi (2008) evaluates the performance of the forecasts with different lengths using data from 2002-2005. All forecast lengths have a similar negative bias of -0.1 m, while the RMS-error increases from 0.3 to almost 0.6 m between the 6 h and 54 h forecast length. Tuomi et al. (2011) verified a six year hindcast (2001-2007) that was forced by winds from FMI's operational HIRLAM. The bias at the NBP wave buoy was -0.1 m and the RMS-error 0.3 m.

The overall performance of the wave model is well documented and we can conclude that smaller wave heights are generally well predicted, especially with the new higher resolution implementation and for the shorter forecasts.

*"In the same idea, I also want to make the observation that some proposals to improve the forecast system could have been welcome as an opening in the conclusion of paragraph 5. "*

We will add a short discussion on this, in the the line of:

"A discussion concerning the issuing of wave warnings for the Baltic Sea should be initiated between the relevant institutes and end users. In addition to re-establishing and harmonising the thresholds of significant wave heights, the use of other parameters (e.g. duration) should also be explored in light of the difficulties of predicting a single maximum value for the wave height. Any decision to include new parameters should be based on the needs of the seafarers. On a more general note, the use of ensemble forecasts might prove useful when issuing wave warnings. An in-depth study is nevertheless needed to quantify to which extent the added information warrants the increased computational cost."

*"In the forecasting paragraph, the comparison between model results and measurements could have been improved by the use of objective indicators (Nash criteria? RMSE?)."*

This is a fair point. Since we wanted this brief communication to focus on the highest wave events, we did not present objective validation indicators for the entire years. These have, however, been calculated by the studies mentioned above. To make our results more objectively comparable with possible future studies into modelling extreme wave conditions, we will add the bias of the model for the 6 m exceedance time to Sections 4.1 and 4.2 (see below):

**Sect. 4.1**: The model bias for the 6 m exceedance time ranged from -0.5 m to -0.8 m in the different forecasts.

**Sect. 4.2**: The model bias for the 6 m exceedance time ranged from -0.7 m to -1.1 m in the different forecasts.

*"The forecast models are compared with a single station for wave parameters. Is this station fully representative of the heterogeneity of the Baltic Sea waves? This point should have been discussed. "*

This is a very good point. The answer to the question is, of course, "no". One point cannot capture the variability of the wave field in the Baltic Proper. We added a short discussion about this to the end of Section 3 so that our research will be easier to put into context:

The wave observations from the NBP cannot be considered entirely representable for the entire Baltic Proper. The highest modelled wave events have been placed either south-southeast of the wave buoy during Gudrun in 2005 (Soomere et al., 2008), slightly west of the wave buoy during Toini in 2017 (Fig. 1.), or slightly east of the wave buoy during Rafael in 2004 (not shown). High waves have also been modelled in the southern Baltic Sea (e.g. Jönsson et al., 2003), which is an area suffering from an acute lack of wave measurements. However, the sparsity of remotely sensed wave data and the uncertainties related to modelling the wave extremes (Fig. 2) underlines the usability of the reliable long term wave buoy measurements presented in this paper.

**New references**

Jönsson, A., Broman, B., and Rahm, L.: Variations in the Baltic Sea wave fields, Ocean Eng., 30, 107 – 126, doi:http://dx.doi.org/10.1016/S0029-8018(01)00103-2, 2003.

Tuomi, L.: The accuracy of FIMR wave forecasts in 2002-2005, MERI – Report Series of the Finnish Institute of Marine Research, 63, 7–17, 2008.

Räämet, A. and Soomere, T.: The wave climate and its seasonal variability in the northeastern Baltic Sea, Estonian J. Earth Sci., 59(1), 100–113, doi:10.3176/earth.2010.1.08, 2010.

Eerola, K.: Twenty-One Years of Verification from the HIRLAM NWP System, Weather and Forecasting, 28, 270–285, doi:10.1175/WAF-D-12-00068.1, 2013.